# Nonlinear Effects of Pulsed Ion Beam in Ultra-High Resolution Material Removal

**DOI:** 10.3390/mi13071097

**Published:** 2022-07-12

**Authors:** Lingbo Xie, Ye Tian, Feng Shi, Ci Song, Guipeng Tie, Gang Zhou, Jianda Shao, Shijie Liu

**Affiliations:** 1College of Intelligence Science and Technology, National University of Defense Technology, Changsha 410073, China; lingbotse@163.com (L.X.); shifeng@nudt.edu.cn (F.S.); sunicris@163.com (C.S.); tieguipeng@163.com (G.T.); zg2206553079@foxmail.com (G.Z.); 2Hunan Key Laboratory of Ultra-Precision Machining Technology, Changsha 410073, China; 3Laboratory of Science and Technology on Integrated Logistics Support, National University of Defense Technology, Changsha 410073, China; 4Laboratory of Thin Film Optics, Shanghai Institute of Optics and Fine Mechanics, Chinese Academy of Sciences, Shanghai 201800, China; jdshao@siom.ac.cn (J.S.); shijieliu@siom.ac.cn (S.L.)

**Keywords:** pulsed ion beam, ultra-high removal resolution, nonlinear effect of ion sputtering

## Abstract

Ion beam sputtering is widely utilized in the area of ultra-high precision fabrication, coating, and discovering the microworld. A pulsed ion beam (PIB) can achieve higher material removal resolution while maintaining traditional ion beam removal performance and macro removal efficiency. In this paper, a 0.01 s pulse width beam is used to sputter atom layer deposition (ALD) coated samples. The nano-scale phenomenon is observed by high-resolution TEM. The results show that when the cumulative sputtering time is less than 1.7 s, the sputtering removal of solid by ion beam is accompanied by a nonlinear effect. Furthermore, the shortest time (0.05 s) and lowest thickness (0.35 nm) necessary to remove a uniform layer of material were established. The definition of its nonlinear effect under a very small removal amount guides industrial ultra-high precision machining. It reveals that PIB not only has high removal resolution on nanoscale, but can also realize high volume removal efficiency and large processing diameter at the same time. These features make PIB promising in the manufacturing of high power/energy laser optics, lithography objective lens, MEMS, and other ultra-high precision elements.

## 1. Introduction

Ultra-precision machining technology with a smaller material resolution is the foundational tool of science and technology and has led the developing direction of modern manufacturing technology [1,2]. The most widely utilized ultra-high resolution processing technologies are nano-optical tweezers, atom force microscope (AFM), ion beam lithography, focused ion beam (FIB), and others [3,4]. Optical tweezers technology uses mechanical action generated by momentum transfer between light and material particles to control the spatial placement of microscopic objects, such as three-dimensional high-precision capture, movement, and arrangement [5,6]. By applying force to its nano-scale scanning probe, AFM achieves atomic writing on the substrate surface. Focused ion beam (FIB) assisted nanolithography technology realizes the removal of nano materials by sputtering focused ions or electrons onto the substrate [7,8]. The above nano-size processing technologies are only suitable for the removal of materials at the scientific research and experimental level due to their low efficiency, they cannot fulfill the criterion of minimal removal at the macro level [9].

Traditional continuous ion beam processing achieves controlled material removal by adjusting the ion beam’s residence time on the substrate surface [10]. Although its removal resolution can theoretically attain sub-nanometer accuracy, it must accurately manage its residence time. As a result, it places great demands on the machine tool’s dynamic performance.

Based on the present state of ultra-precision machining, our team [11] and colleagues [10,12] control the ion source using a pulse power supply. Based on the original time-domain control, frequency-domain parameters are added to discretize the typical continuous ion beam into a pulse beam with configurable pulse width and frequency. It not only offers ultra-high removal resolution, but also decreases the need for the machine tool’s dynamic performance and eliminates the formation of extra removal layers.

We conducted a gradient pulse experiment earlier with a pulse width of 0.1 milliseconds (ms) and a cumulative sputtering duration of 1.2–4.8 s to validate the linear connection between the removal depth and the number of pulses when a pulsed ion beam is sputtered with a large number of pulses. The removal quantity of a single pulse with a pulse width of 0.1 ms (6.7 × 10^−4^ nm) is calculated using an analogy.

However, according to Rodolfo et al. [13,14], when a low-energy ion beam sputters on a solid surface, the sputtering yield is not always stable, thus it is worth examining the method of determining the limit removal resolution of the PIB by averaging accumulatively. When removal levels approach several hundred picometers, it is unclear whether stable and predictable material removal efficiency still exists or not [15,16,17].

For the mechanism of ion sputtering removal, Sigmund [18] separates the interaction between accelerated ions and solids into three stages, which are mostly governed by the energy of the incoming ions: single-knock-on regime, linear cascade regime, and spike regime. It is commonly regarded as a “linear cascade” mode for solid sputtering with input ion energies ranging from 1 keV to 30 keV [19].

In an article published in Nature, Jiali Li et al. [20] described a similar event to Rodolfo’s finding. They attempted to create molecular or nanopores in thin insulating solid sheets. During the study process, they discovered that the atoms on the material’s surface will “flow” into the pores when subjected to the action of an ion beam. However, due to a lack of in-depth investigation, their studies have merely conjectured and described these events.

Yuriy et al. [21] proposed a thermal spot sputtering mode between linear cascade and spike. The model applies to some existing experimental results that cannot be explained by linear cascade and single knock-on sputtering models. They used hydrodynamic to simulate the sputtering process of various metals by cesium ions with energies ranging from 1 keV to 30 keV and found that there was a formation of “quasi liquid” or “melt” in the sputtering process. The nonlinear cascade mode opposite to the linear cascade mode will appear [22,23]. The “melt” theory can well explain the research results of Rodolfo and Jiali Li et al.

In the second section, we describe the mechanism of low-energy ion beam sputtering proposed by Yuriy et al. and try to compare it to the sputtering removal method of PIB. When the cumulative sputtering time is short enough (<1.7 s), the total energy of incident ions can be considered to be the same as that in the case of low-energy ion beam sputtering. Furthermore, in the third section, we completed the experiment of short-time pulsed ion sputtering. The results show that in the above cases, the material removal depth is nonlinearly related to the cumulative sputtering time of the pulsed ion beam. This phenomenon has substantial implications for the novel atomic-level material removal processing approach.

## 2. “Thermal Spot” Model Theory

In this part, we calculated the ion energy emitted by the pulsed ion beam. According to the relevant research on the characteristics of single-hole beam current, the ions emitted by the ion sheath are extracted into the ion beam after passing through the screen grid and obey the law of the three-second power due to the limitation of space charge [24]. The beam density of a single hole can be calculated by Equation (1). The corresponding meanings of all symbol abbreviations are given in the Abbreviation.
(1)$$\begin{array}{c} J=\frac{4}{9}\varepsilon_{0}\sqrt{\frac{2e}{M_{i}}}\frac{V^{\frac{3}{2}}}{L_{g}^{2}} \end{array}$$

In Equation (1): *V* represents the voltage between the screen grid and the accelerator grid, $L_{g}$ represents the distance between the screen grid and the accelerator grid, $M_{i}\;$ represents the ion mass, e  represents the charge and electric quantity, and represents the vacuum dielectric constant. To make the design of the ion optical system closer to the ideal bipolar plate model, Dr. Kaufman [25] modified the above formula, using effective acceleration length $L_{e}=\sqrt{L_{g}^{2}+\frac{d_{s}^{2}}{4}}$ instead of $L_{g}$ and total acceleration voltage $V_{t}$ instead of *V*. Assuming that the ion beam from the gate hole is uniform, the beam intensity *I* from the single hole can be obtained by multiplying the beam density of the single hole by the area of the small hole.
(2)$$\begin{array}{c} I=\frac{\pi d_{s}^{2}}{4}J=\frac{\pi }{36}\varepsilon_{0}\sqrt{\frac{2e}{M_{i}}}\frac{V_{t}^{\frac{3}{2}}d_{s}^{2}}{L_{e}^{2}} \end{array}$$

Ejected energy *E*:
(3)$$\begin{array}{c} E=\frac{1}{f}DC\cdot t\cdot I \end{array}$$

In Equation (3), *f* is pulse frequency, *DC* is pulse duty factor and *t* is ion beam sputtering time. When the frequency and duty factor are determined, the ion beam emission energy can be controlled by controlling the ion beam sputtering time.

Yuriy et al. proposed a model of ultra-low energy ion sputtering based on cascade volume effect (i.e., thermal spot model) for the phenomena presented in the first section. The cesium ion sputtering experiment with 250–1000 eV energy shows that the actual cesium ion concentration detected on the sputtering surface is the same as that calculated in the formula, which verifies that the low energy ion sputtering yield conforms to the hot spot model. In this model, the formation of quasi-liquid or melt is considered during the sputtering process, resulting in a non-linear cascade which is opposite to the linear cascade mode. The sputtering yield is determined by both the linear cascade sputtering and the non-linear thermal sputtering $Y_{e}$:(4)$$\begin{array}{c} Y=Y_{b}+Y_{e} \end{array}$$

According to Sigmund sputtering theory [18], in the linear cascade stage ($Y_{b}$), the sputtering yield is proportional to the energy deposited by the incident ions on the solid surface:(5)$$Y_{b}=\frac{0.042}{U_{0}}\left(0.15+0.13\frac{M_{2}}{M_{1}}\right)s_{n}\left(E\right)$$
where $s_{n}\left(E\right)$ can be obtained by integral of the Equation (3). During the thermal sputtering stages ($Y_{e}$), the evaporation rate Φ per unit time and surface area given by:(6)$$\Phi =N{\left(kT/2\pi M_{2}\right)}^{\frac{1}{2}}exp\left(-U_{0}∕kT\right)$$

*N* is the atomic density of the sample material, $M_{2}$ is the mass of the target atoms and $U_{0}$ is the binding energy of the sample surface. With *k* being Boltzmann’s constant, the surface temperature *T* given by:(7)$$T=\frac{2}{3Nk}\cdot \frac{s_{n}\left(E\right)}{\pi \varrho^{2}}$$

$\pi \varrho^{2}$ being the hot surface area, and the sputtering yield:(8)$$Y_{e}=\pi \varrho^{2}\cdot \Phi$$

By integrating the constant terms, Equation (4) can be simplified to:(9)y=At+Bexp(Ct)+D

According to Yuriy’s research conclusion, when the incident ion energy is low, the contribution percentage of “quasi-liquid” or “melt” to the sputtering yield is much higher than that of the incident ion energy [26,27].

Consequently, when the incident ion energy is high, $Y_{b}$ is much larger than $Y_{e}$, and the melt $Y_{e}$ formed by the sputtering can be neglected compared with the material Y removed. So, the sputtering yield is linear with the incident ion energy. However, when the incident ion energy is low, the material sputtering removal Y is small and the difference between $Y_{e}$ and $Y_{b}$ is not significant. Compared with the material Y removed by sputtering, the melt $Y_{e}$ formed by sputtering cannot be ignored but has a greater influence, which makes the non-linear effect appear.

The incidence energy of low quantities of pulsed ions in PIB is within the thermal spot model’s applicable range, therefore it should have a non-linear effect.

We aimed to test the short-time pulsed ion sputtering removal rule in order to accomplish controlled ultra-high resolution material removal of PIB. The film was sputtered by PIB with varying sputtering times after an ALD coating was plated on the surface of an ultra-smooth (Ra < 0.2 nm) fused quartz sample. Finally, the removed film was subjected to high resolution transmission electron microscope (HRTEM). The film depth is observed after sputtering, and the ion sputtering removal rule with low pulse number was determined.

## 3. Sputtering Experiments and Phenomena Analysis

### 3.1. Removal Depth Rule

In order to calculate the removal amount by comparing the film layer with the substrate and the film layer before processing.

Five ultra-smooth (Ra < 0.5 nm) fused quartz samples with 22 mm diameter and 6 mm thickness were filmed by ALD. The film material is hafnium oxide, and the thickness uniformity is less than ±0.1%.

The theoretical removal amount of hafnium oxide film is calculated according to Equation (9). Considering the removal amount of single pulse ion sputtering and the observation effect after processing, the coating thickness is set at 5 nm and the samples are coated in the same batch to ensure uniform thickness between samples.

After coating, HRTEM test is carried out on the sample. The information of initial surface atomic structure and film thickness is shown in Figure 1a,b. The uniformity of the film is below 0.1 nm. Better film uniformity is convenient for error elimination and correction in subsequent experiments.

Five samples were removed by ion beam pulsed sputtering with the parameters shown in Table 1, of which the waveform of the five pulsed beams is shown in Figure 1e.

The fundamental mechanism of physical sputtering is the momentum exchange between ions and material atoms, and the splashed particles are mainly neutral atoms of the material. These neutral atoms are in the excited state at the beginning of sputtering and return to the ground state after about 10^−7^ s. Therefore, the pulse interval of 1 Hz is much greater than its relaxation time. Under the parameters in Table 1, the sputtering time of single pulse is 0.01 s. After converting the different pulses in Figure 1 into the accumulated sputtering time, the corresponding removal amount is shown in Table 2. The three groups of data are indicated in the table.

The total removal amount grows with the increase of the cumulative sputtering time, but increases abruptly at 0.2 s, and the trend of the removal amount becomes routinely non-linear with the growth of the sputtering time afterwards, as shown by the actual removal line in Figure 2. Thus, when the cumulative sputtering time is less than 0.2 s, the energy of ion beam pulse sputtering on the coating surface does not reach the threshold value at which the linear cascade splashing components begin to generate in the “thermal spot” model (i.e., the $Y_{b}$ term in Equation (4) is 0 and the sputtering yield is only $Y_{e}$ of thermal sputtering). When the accumulated sputtering time is increased to 0.2 s, the linear cascade sputtering begins and the sputtering yield is $Y_{e}$ adding $Y_{b}$, which shows a sharp increase in the removal amount in Figure 2.

### 3.2. Sputtering Model Transition Point

After obtaining the non-linear rule and model of low energy PIB sputtering as shown in Figure 2. We are interested in the exact value of the mutation point (i.e., the shortest time for hafnium oxide material to reach the linear cascade sputtering threshold and the minimum thickness of the material to be removed under the parameter of Table 1). This physical quantity has a high reference value for material atomic layer removal. So, we experimented with this as well.

The period in which the material is initially eliminated is 0.1–0.2 s, which corresponds to the number of pulses 10–20 times in the findings presented in Figure 2. However, the thickness of the coating removed between 0.05–0.1 s changes little, implying that the sputtering yield is still driven solely by $Y_{e}$. As a result, we divided the pulse number interval into five gradients (cumulative sputtering time is 0.05 s). Samples with an initial coating thickness of 5 nm are experimented using the pulse settings listed in Table 1. The processing detection region corresponds to the one shown in Figure 1. The coating is HRTEM observed after the pulse sputtering test. Figure 3 shows that the removal thickness is 0.494 nm after 15 pulses (cumulative sputtering time is 0.15 s). The correlation between removal amount (below 0.2 s) and sputtering time is shown in Figure 4. The removal amount increases slowly from 0.05 s to 0.15 s and increases sharply at 0.2 s.

As a consequence of the results in Figure 4, it can be deduced that when the cumulative sputtering duration reaches 0.2 s, the components of sputtering yield shift from single thermal sputtering to linear superimposed heating sputtering mode (i.e., thermal spot mode). As seen in Figure 5, the sputtering yield is overlaid by the $Y_{b}$ and $Y_{e}$ terms.

## 4. Discussion

PIB retains the benefits of classic continuous ion beam processing, as well as the better controllability of pulsed laser processing and the same or superior material removal ability as FIB. By regulating the number of pulses, pulse duty cycle, and pulse frequency employed in PIB processing, we may achieve atomic resolution removal in the square centimeter region, and as a result the maximum removal aperture is significantly bigger than FIB. This flexible large-aperture ultra-high resolution removal method has several applications, including harmonic oscillator fine-quality tuning, ultra-precision optical component manufacturing, micro-nano electronics technology, and so on.

In this paper, we discussed the non-linear effect of low energy ion sputtering on the sputtering mechanism and compared it to Zhou’s ion beam removal rule. We discovered a non-linear effect in the removal of low-quantity pulsed ion beam sputtering. The experiment of sputtering removal of the ALD coating layer was carried out by setting up pulse gradient to vary the cumulative sputtering duration of ion beam, and the non-linear law and mathematical model of low energy pulse ion beam sputtering were established. This model is based on certain characteristics (Table 1) and materials (amorphous hafnium oxide), but it does not demonstrate the model’s limits. According to the relationship between thermal sublimation coefficient and surface atomic binding energy of amorphous hafnium oxide and required materials, the conversion coefficient can be derived for different materials, which means that the model can be modified by the conversion coefficient between different materials.

In Section 3, we acquire the least removal thickness of 0.35 nm, which is just obtained for the convenience of observing experimental phenomena (***f*** = 1 Hz), but the minimum controllable removal resolution can be achieved by increasing the frequency of the pulse. The length of a single pulse lowers as the frequency increases, and the removal resolution multiplies and the removal efficiency improves as the duty cycle grows. These controllable conditions make PIB a significant basic tool in the field of nanoscience, allowing for deterministic material removal at atomic level resolution, indicating that ion-beam-based processing technology has progressed from the analog to the digital age.

## 5. Conclusions

Clear material removal rule of PIB has guiding significance to actual processing. Through the research on non-linear mechanism of low energy ion sputtering, the removal rule should be non-linear under conditions of low quantity of impulses by analogy. This deduction was verified by the experiments carried out.

Sputtering removal of ALD coated samples is carried out by means of gradient quantity pulse and HRTEM is used to observe the sputtered samples. The result shows that there is a non-linear effect on the PIB sputtering removal yield below 170 times. The material sputtering removal amount is exponentially related to the pulse number and a material removal model suitable for this stage is obtained. It is also confirmed that the minimum number of pulses required to accomplish uniform material removal is 5 and the minimum removal depth is 0.35 nm within the application range of the model. This research result may have potential significance to the further development of ultra-precision machining and the realization of controllable atom-level manufacturing. It has great potential in the fields of microelectronics, electronic information and materials, such as harmonic oscillator micro mass adjustment, large-area microstructure preparation, nano scale field effect transistor (FET) manufacturing, etc.

## Figures and Tables

**Figure 1 micromachines-13-01097-f001:**
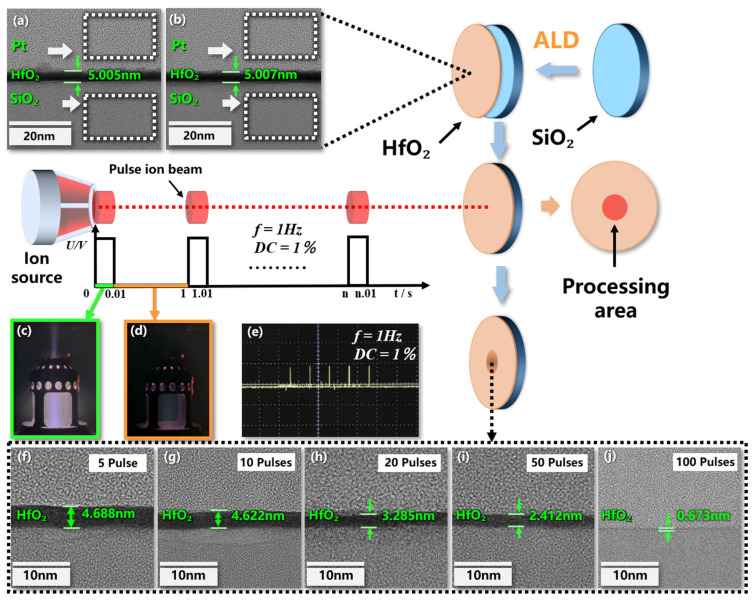
Changes of coating layers during pulsed ion beam processing: (**a**,**b**) Initial coating information of random two samples; (**c**) beam on; (**d**) beam off; (**e**)pulse waveform; (**f**) after 5 pulses; (**g**) after 10 pulses; (**h**) after 20 pulses; (**i**) after 50 pulses; (**j**) after 100 pulses.

**Figure 2 micromachines-13-01097-f002:**
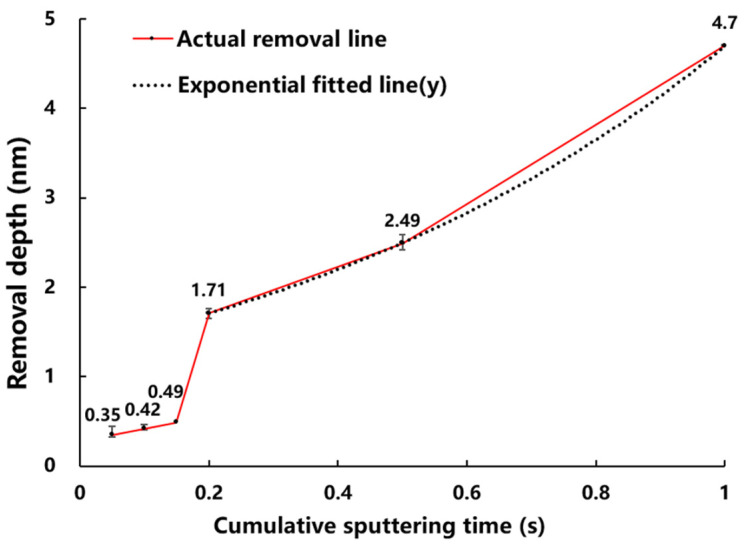
Removal depth versus cumulative sputtering time.

**Figure 3 micromachines-13-01097-f003:**
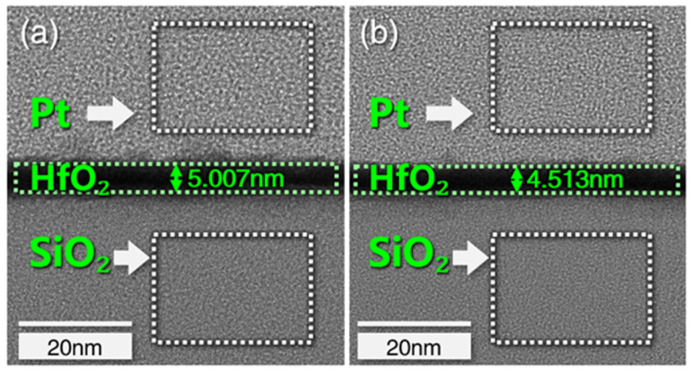
(**a**) Initial coating section; (**b**) after 15 pulses.

**Figure 4 micromachines-13-01097-f004:**
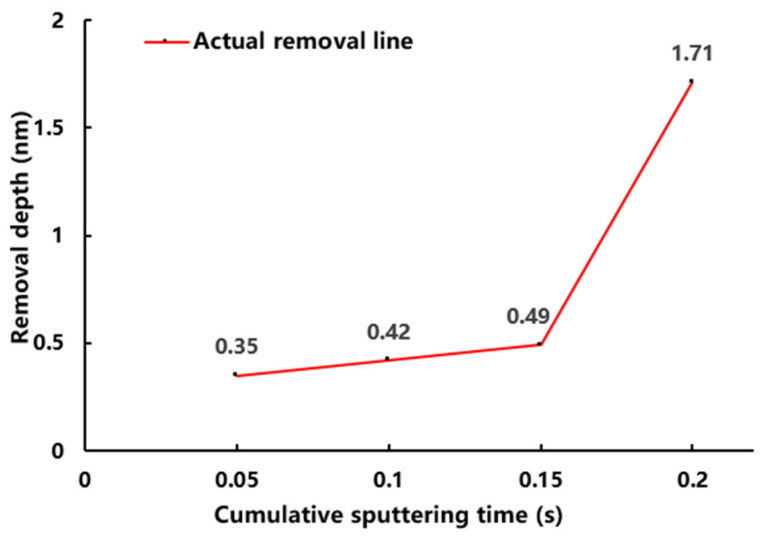
Relationship between removal depth and accumulated sputtering time (below 0.2 s).

**Figure 5 micromachines-13-01097-f005:**
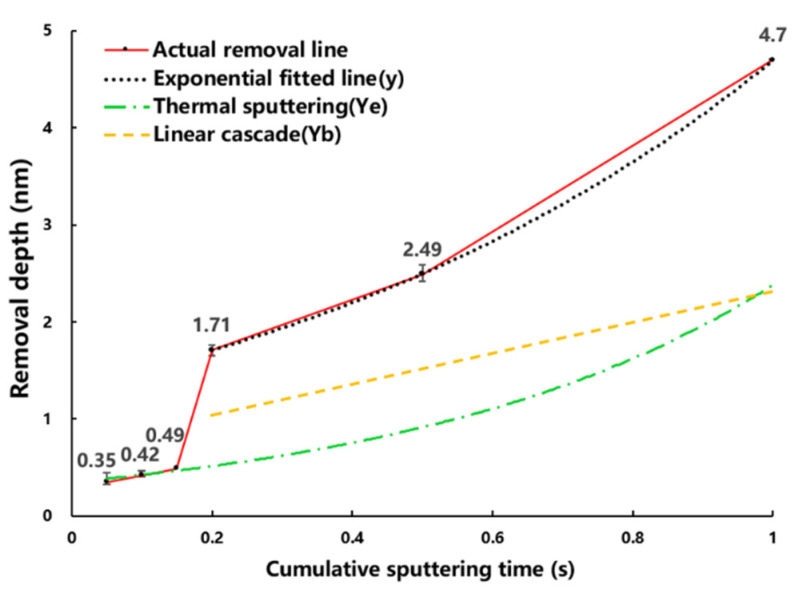
Linear cascade superimposed heating sputtering in thermal spot mode.

**Table 1 micromachines-13-01097-t001:** PIB processing parameters.

Parameter	Value	Parameter	Value
Ion energy	600 eV	Beam diameter	10 mm
Frequency	1 Hz	Pulse length	10 ms
Ion Species	Ar+	Sputtering angle	90°

**Table 2 micromachines-13-01097-t002:** Removal depths at different cumulative sputtering times.

**Sputtering Times (s)**	0.05	0.1	0.2	0.5	1
**Removal Depths (nm)**	0.33	0.39	1.65	2.42	4.68
	0.35	0.42	1.71	2.49	4.7
	0.45	0.46	1.76	2.59	4.73

## Data Availability

The data presented in this study are available on request from the corresponding author. The data are not publicly available due to the data also forming part of an ongoing study.

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
