# Peer review of "Nonlinear Effects of Pulsed Ion Beam in Ultra-High Resolution Material Removal"

_micromachines, 2022, doi:10.3390/mi13071097_

Round 1
Reviewer 1 Report
The authors has proposed a fast process to sputter atom layer deposition ALD. The paper fits the scope of the journal and needs minor revision before it gets published. Here is a list of the concerns:
- .“. Thermal spot” model theory (please correct typo)
- Extra space in line 113
- Add a list of abbreviations to define each symbol in the text
- Section 3 needs a major revision
- How long it takes to remove a thin layer during this process?
- Typo in line 174
- Legend and axis title in Figure 2 are not clear.
- Figure 2 shows almost a linear response. Where is the nonlinearity that you are mentioned in the text?
- Extra space in 205
- The paper is discussing a novel idea but the way it presents the work needs to be improved.
I highly recommend it for publication after it gets revised and modified.
Author Response
Response to Reviewer 1 Comments
Point 1: .“. Thermal spot” model theory (please correct typo).
Response 1: The corresponding part has been modified as suggested.
Point 2: Extra space in line 113.
Response 2: The corresponding part has been modified as suggested.
Point 3: Add a list of abbreviations to define each symbol in the text.
Response 3: There are many symbols in the text, which are not suitable to be given directly in the manuscript. It has been given in the appendix now.
Point 4: Section 3 needs a major revision.
Response 4: The corresponding part has been modified as suggested.
Point 5: How long it takes to remove a thin layer during this process?
Response 5: As shown in Table 2, We convert the number of pulses in Figure 1 to the corresponding sputtering time. The number of pulses in each group was sampled three times and the removal depth of the film was also given in Table 2 (eg: Accumulated sputtering for 0.05 seconds with average removal of 0.37nm).
Point 6: Typo in line 174.
Response 6: The corresponding part has been modified as suggested.
Point 7: Legend and axis title in Figure 2 are not clear.
Response 7: The corresponding part has been modified as suggested.
Point 8: Figure 2 shows almost a linear response. Where is the nonlinearity that you are mentioned in the text?
Response 8: The non-linear effect of sputter removal and pulse number mainly occurs in the low number area. It can be seen from Figure 2 that there is a significant change in the removal amount at 20 times. Further validation has also been carried out and the results are shown in Figure 4. The removal amount before and after this sudden change point (20times) indicates that the removal amount and pulse number are non-linear at low number of pulses(<170times).
Point 9: Extra space in 205
Response 9: The corresponding part has been modified as suggested.
Point 10: The paper is discussing a novel idea but the way it presents the work needs to be improved.
Response 10: The phenomenon we found is novel, and the research on it is in the initial stage. We will improve the research method in the next step.

Reviewer 2 Report
The paper entitled " Nonlinear Effects of Pulsed Ion Beam in Ultra-High Resolution 2 Material Removal" by Xie et al. shows the investigation of sputtering effect of pulsed ion beam on thin coating films. They studied the non-linearity of the sputtering yield as a function of the cumulative time of sputtering in low energy range. The proposed manuscript shows clear experimental results as well as strong theoretical background.
I feel that the paper should be accepted in its present form.
Author Response
Thank you for your review!
Wish everything goes well with your work!